## META-RESEARCH

# Tracking the popularity and outcomes of all bioRxiv preprints

**Abstract** The growth of preprints in the life sciences has been reported widely and is driving policy changes for journals and funders, but little quantitative information has been published about preprint usage. Here, we report how we collected and analyzed data on all 37,648 preprints uploaded to bioRxiv.org, the largest biology-focused preprint server, in its first five years. The rate of preprint uploads to bioRxiv continues to grow (exceeding 2,100 in October 2018), as does the number of downloads (1.1 million in October 2018). We also find that two-thirds of preprints posted before 2017 were later published in peer-reviewed journals, and find a relationship between the number of downloads a preprint has received and the impact factor of the journal in which it is published. We also describe Rxivist.org, a web application that provides multiple ways to interact with preprint metadata.
DOI: https://doi.org/10.7554/eLife.45133.001

**RICHARD J ABDILL AND RAN BLEKHMAN\***

## Introduction

In the 30 days of September 2018, four leading biology journals – *The Journal of Biochemistry*, *PLOS Biology*, *Genetics* and *Cell* – published 85 full-length research articles. The preprint server bioRxiv (pronounced 'Bio Archive') had posted this number of preprints by the end of September 3 (*Figure 1—source data 4*). Preprints allow researchers to make their results available as quickly and widely as possible, short-circuiting the delays and requests for extra experiments often associated with peer review (*Berg et al., 2016*; *Powell, 2016*; *Raff et al., 2008*; *Snyder, 2013*; *Hartgerink, 2015*; *Vale, 2015*; *Royle, 2014*).

Physicists have been sharing preprints using the service now called arXiv.org since 1991 (*Verma, 2017*), but early efforts to facilitate preprints in the life sciences failed to gain traction (*Cobb, 2017*; *Desjardins-Proulx et al., 2013*). An early proposal to host preprints on PubMed Central (*Varmus, 1999*; *Smaglik, 1999*) was scuttled by the National Academy of Sciences, which successfully negotiated to exclude work

that had not been peer-reviewed (*Marshall, 1999*; *Kling et al., 2003*). Further attempts to circulate biology preprints, such as NetPrints (*Delamothe et al., 1999*), Nature Precedings (*Kaiser, 2017*), and The Lancet Electronic Research Archive (*McConnell and Horton, 1999*), popped up (and then folded) over time (*The Lancet Electronic Research Archive, 2005*). The preprint server that would catch on, bioRxiv, was not founded until 2013 (*Callaway, 2013*). Now, biology publishers are actively trawling preprint servers for submissions (*Barsh et al., 2016*; *Vence, 2017*), and more than 100 journals accept submissions directly from the bioRxiv website (*BioRxiv, 2018*). The National Institutes of Health now allows researchers to cite preprints in grant proposals (*National Institutes of Health, 2017*), and grants from the Chan Zuckerberg Initiative require researchers to post their manuscripts to preprint servers (*Chan Zuckerberg Initiative, 2019*; *Champieux, 2018*).

Preprints are influencing publishing conventions in the life sciences, but many details about

\*For correspondence: blekhman@ umn.edu

**Competing interests:** The authors declare that no competing interests exist.

the preprint ecosystem remain unclear. We know bioRxiv is the largest of the biology pre-print servers (*Anaya, 2018*), and sporadic updates from bioRxiv leaders show steadily increasing submission numbers (*Sever, 2018*). Analyses have examined metrics such as total downloads (*Serghiou and Ioannidis, 2018*) and publication rate (*Schloss, 2017*), but long-term questions remain open. Which fields have posted the most preprints, and which collections are growing most quickly? How many times have preprints been downloaded, and which catego-ries are most popular with readers? How many preprints are eventually published elsewhere, and in what journals? Is there a relationship between a preprint's popularity and the journal in which it later appears? Do these conclusions change over time?

Here, we aim to answer these questions by collecting metadata about all 37,648 preprints posted to bioRxiv from its launch through November 2018. As part of this effort we have developed Rxivist (pronounced 'Archivist'): a website, API and database (available at https://rxivist.org and gopher://origin.rxivist.org) that provide a fully featured system for interacting programmatically with the periodically indexed metadata of all preprints posted to bioRxiv.

## Results

We developed a Python-based web crawler to visit every content page on the bioRxiv website and download basic data about each preprint across the site's 27 subject-specific categories: title, authors, download statistics, submission date, category, DOI, and abstract. The bioRxiv website also provides the email address and institutional affiliation of each author, plus, if the preprint has been published, its new DOI and the journal in which it appeared. For those pre-prints, we also used information from Crossref to determine the date of publication. We have stored these data in a PostgreSQL database; snapshots of the database are available for download, and users can access data for individual preprints and authors on the Rxivist website and API. Additionally, a repository is available online at https://doi.org/10.5281/zenodo.2465689 that includes the database snapshot used for this manuscript, plus the data files used to create all figures. Code to regenerate all the figures in this paper is included there and on GitHub (https://github.com/blekhmanlab/rxivist/

blob/master/paper/figures.md). See Methods and Supplementary Information for a complete description.

### Preprint submissions

The most apparent trend that can be pulled from the bioRxiv data is that the website is becoming an increasingly popular venue for authors to share their work, at a rate that increases almost monthly. There were 37,648 preprints available on bioRxiv at the end of November 2018, and more preprints were posted in the first 11 months of 2018 (18,825) than in all four previous years combined (*Figure 1a*). The number of bioRxiv preprints doubled in less than a year, and new submis-sions have been trending upward for five years (*Figure 1b*). The largest driver of site-wide growth has been the neuroscience collection, which has had more submissions than any bio-Rxiv category in every month since September 2016 (*Figure 1b*). In October 2018, it became the first of bioRxiv's collections to contain 6,000 preprints (*Figure 1a*). The second-largest cate-gory is bioinformatics (4,249 preprints), followed by evolutionary biology (2,934). October 2018 was also the first month in which bioRxiv posted more than 2,000 preprints, increasing its total preprint count by 6.3% (2,119) in 31 days.

### Preprint downloads

Using preprint downloads as a metric for reader-ship, we find that bioRxiv's usage among read-ers is also increasing rapidly (*Figure 2*). The total download count in October 2018 (1,140,296) was an 82% increase over October 2017, which itself was a 115% increase over October 2016 (*Figure 2a*). BioRxiv preprints were downloaded almost 9.3 million times in the first 11 months of 2018, and in October and November 2018, bio-Rxiv recorded more downloads (2,248,652) than in the website's first two and a half years (*Figure 2b*). The overall median downloads per paper is 279 (*Figure 2b*, inset), and the geno-mics category has the highest median down-loads per paper, with 496 (*Figure 2c*). The neuroscience category has the most downloads overall – it overtook bioinformatics in that metric in October 2018, after bioinformatics spent nearly four and a half years as the most down-loaded category (*Figure 2d*). In total, bioRxiv preprints were downloaded 19,699,115 times from November 2013 through November 2018,

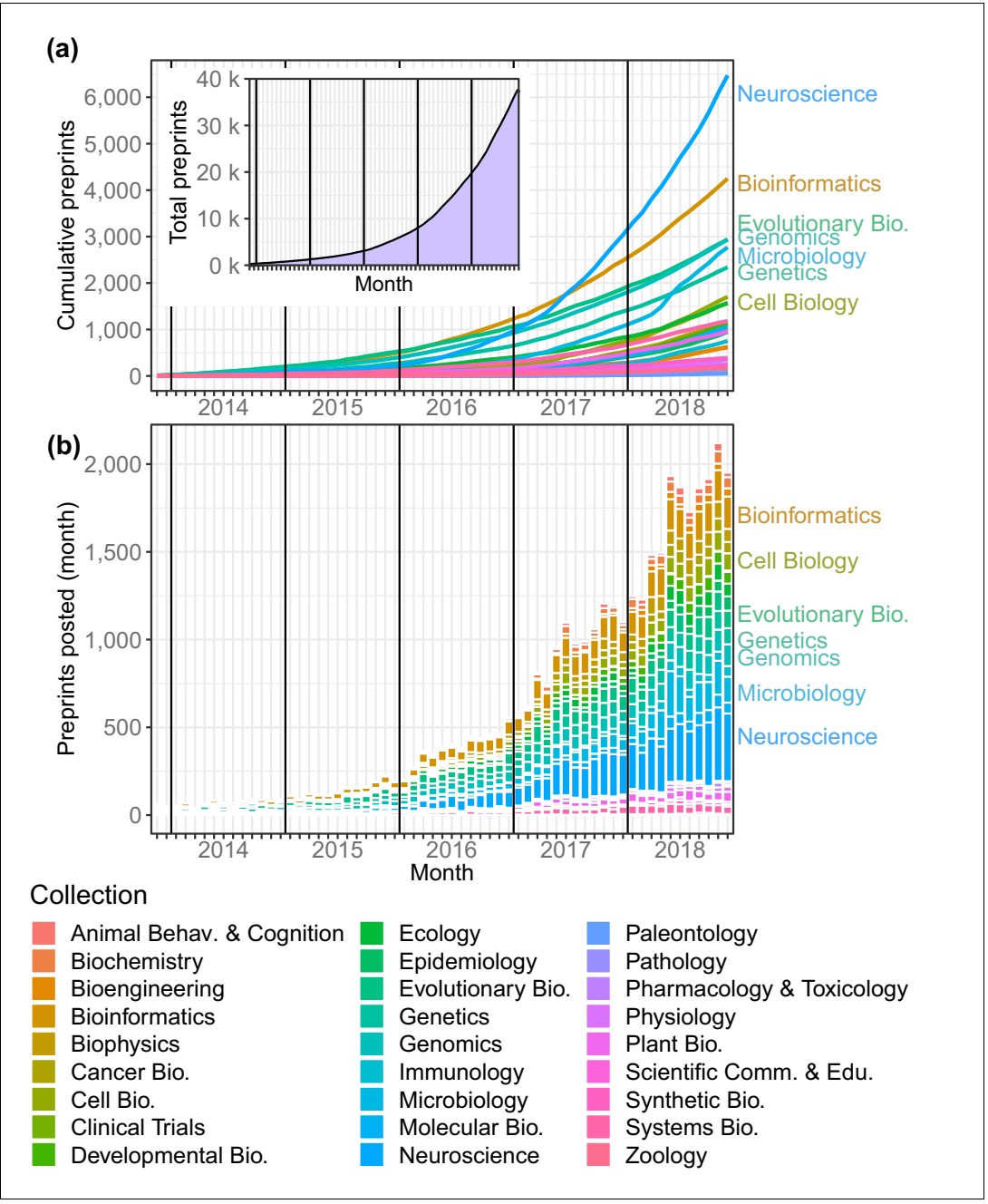

**Figure 1.** Total preprints posted to bioRxiv over a 61 month period from November 2013 through November 2018. (a) The number of preprints (y-axis) at each month (x-axis), with each category depicted as a line in a different color. Inset: The overall number of preprints on bioRxiv in each month. (b) The number of preprints posted (y-axis) in each month (x-axis) by category. The category color key is provided below the figure.
DOI: https://doi.org/10.7554/eLife.45133.002

The following source data is available for figure 1:

**Source data 1.** The number of submissions per month to each bioRxiv category, plus running totals.
DOI: https://doi.org/10.7554/eLife.45133.003

**Source data 2.** An Excel workbook demonstrating the formulas used to calculate the running totals in *Figure 1— source data 1*.
DOI: https://doi.org/10.7554/eLife.45133.004

**Source data 3.** The number of submissions per month overall, plus running totals.
DOI: https://doi.org/10.7554/eLife.45133.005

*Figure 1 continued on next page*

*Figure 1 continued*

**Source data 4.** The number of full-length articles published by an arbitrary selection of well-known journals in September 2018.
DOI: https://doi.org/10.7554/eLife.45133.006
**Source data 5.** A table of the top 15 authors with the most preprints on bioRxiv.
DOI: https://doi.org/10.7554/eLife.45133.007
**Source data 6.** A list of every author, the number of preprints for which they are listed as an author, and the number of email addresses they are associated with.
DOI: https://doi.org/10.7554/eLife.45133.008
**Source data 7.** A table of the top 25 institutions with the most authors listing them as their affiliation, and how many papers have been published by those authors.
DOI: https://doi.org/10.7554/eLife.45133.009
**Source data 8.** A list of every indexed institution, the number of authors associated with that institution, and the number of papers authored by those researchers.
DOI: https://doi.org/10.7554/eLife.45133.010

and the neuroscience category's 3,184,456 total downloads accounts for 16.2% of these (*Figure 2d*). However, this is driven mostly by that category's high volume of preprints: the median downloads per paper in the neuroscience category is 269.5, while the median of preprints in all other categories is 281 (*Figure 2c*; Mann–Whitney *U* test p=0.0003).

We also examined traffic numbers for individual preprints relative to the date that they were posted to bioRxiv, which helped create a picture of the change in a preprint's downloads by month (*Figure 2—figure supplement 1*). We can see that preprints typically have the most downloads in their first month, and the download count per month decays most quickly over a preprint's first year on the site. The most downloads recorded in a preprint's first month is 96,047, but the median number of downloads a preprint receives in its debut month on bioRxiv is 73. The median downloads in a preprint's second month falls to 46, and the third month median falls again, to 27. Even so, the average preprint at the end of its first year online is still being downloaded about 12 times per month, and some papers don't have a 'big' month until relatively late, receiving the majority of their downloads in their sixth month or later (*Figure 2—figure supplement 2*).

### Preprint authors

While data about the authors of individual preprints is easy to organize, associating authors between preprints is difficult due to a lack of consistent unique identifiers (see Methods). We chose to define an author as a unique name in the author list, including middle initials but disregarding letter case and punctuation. Keeping this in mind, we find that there are 170,287

individual authors with content on bioRxiv. Of these, 106,231 (62.4%) posted a preprint in 2018, including 84,339 who posted a preprint for the first time (*Table 1*), indicating that total authors increased by more than 98% in 2018.

Even though 129,419 authors (76.0%) are associated with only a single preprint, the mean preprints per author is 1.52 because of a skewed rate of contributions also found in conventional publishing (*Rørstad and Aksnes, 2015*): 10% of authors account for 72.8% of all preprints, and the most prolific researcher on bioRxiv, George Davey Smith, is listed on 97 preprints across seven categories (*Figure 1—source data 5*). 1,473 authors list their most recent affiliation as Stanford University, the most represented institution on bioRxiv (*Figure 1—source data 7*). Though the majority of the top 100 universities (by author count) are based in the United States, five of the top 11 are from Great Britain. These results rely on data provided by authors, however, and is confounded by varying levels of specificity: while 530 authors report their affiliation as 'Harvard University,' for example, there are 528 different institutions that include the phrase 'Harvard,' and the four preprints from the 'Wyss Institute for Biologically Inspired Engineering at Harvard University' don't count toward the 'Harvard University' total.

### Publication outcomes

In addition to monthly download statistics, bioRxiv also records whether a preprint has been published elsewhere, and in what journal. In total, 15,797 bioRxiv preprints have been published, or 42.0% of all preprints on the site (*Figure 3a*), according to bioRxiv's records linking preprints to their external publications. Proportionally, evolutionary biology preprints have

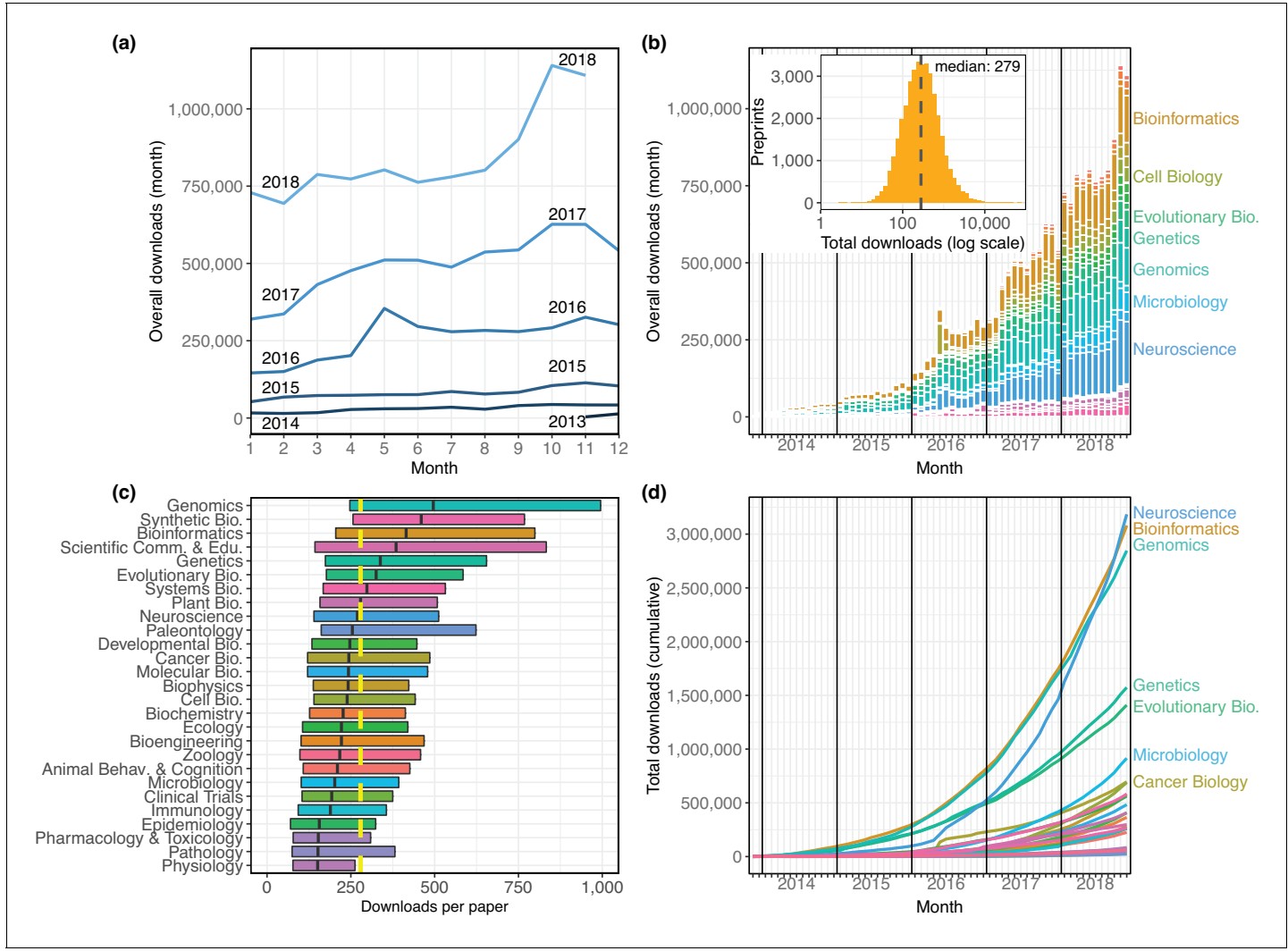

**Figure 2.** The distribution of all recorded downloads of bioRxiv preprints. (a) The downloads recorded in each month, with each line representing a different year. The lines reflect the same totals as the height of the bars in *Figure 2b*. (b) A stacked bar plot of the downloads in each month. The height of each bar indicates the total downloads in that month. Each stacked bar shows the number of downloads in that month attributable to each category; the colors of the bars are described in the legend in *Figure 1*. Inset: A histogram showing the site-wide distribution of downloads per preprint, as of the end of November 2018. The median download count for a single preprint is 279, marked by the yellow dashed line. (c) The distribution of downloads per preprint, broken down by category. Each box illustrates that category's first quartile, median, and third quartile (similar to a boxplot, but whiskers are omitted due to a long right tail in the distribution). The vertical dashed yellow line indicates the overall median downloads for all preprints. (d) Cumulative downloads over time of all preprints in each category. The top seven categories at the end of the plot (November 2018) are labeled using the same category color-coding as above.

DOI: https://doi.org/10.7554/eLife.45133.011

The following source data and figure supplements are available for figure 2:

**Source data 1.** A list of every preprint, its bioRxiv category, and its total downloads.

DOI: https://doi.org/10.7554/eLife.45133.021

**Source data 2.** The number of downloads per month in each bioRxiv category, plus running totals.

DOI: https://doi.org/10.7554/eLife.45133.022

**Source data 3.** An Excel workbook demonstrating the formulas used to calculate the running totals in *Figure 2—source data 2*.

DOI: https://doi.org/10.7554/eLife.45133.023

**Source data 4.** The number of downloads per month overall, plus running totals.

DOI: https://doi.org/10.7554/eLife.45133.024

**Figure supplement 1.** The distribution of downloads that preprints accrue in their first months on bioRxiv.

DOI: https://doi.org/10.7554/eLife.45133.012

**Figure supplement 1—source data 1.** Monthly download counts for each bioRxiv preprint for each of its first 12 months.

*Figure 2 continued on next page*

*Figure 2 continued*

DOI: https://doi.org/10.7554/eLife.45133.013

**Figure supplement 2.** The proportion of downloads that preprints accrue in their first months on bioRxiv.

DOI: https://doi.org/10.7554/eLife.45133.014

**Figure supplement 3.** Multiple perspectives on per-preprint download statistics.

DOI: https://doi.org/10.7554/eLife.45133.015

**Figure supplement 3—source data 1.** The download counts for each bioRxiv preprint in its first month online.

DOI: https://doi.org/10.7554/eLife.45133.016

**Figure supplement 3—source data 2.** Maximum monthly download count for each bioRxiv preprint.

DOI: https://doi.org/10.7554/eLife.45133.017

**Figure supplement 3—source data 3.** A list of each bioRxiv preprint and how many downloads it received in 2018.

DOI: https://doi.org/10.7554/eLife.45133.018

**Figure supplement 4.** Total downloads per preprint, segmented by the year in which each preprint was posted.

DOI: https://doi.org/10.7554/eLife.45133.019

**Figure supplement 4—source data 1.** A list of each bioRxiv preprint and how many downloads it received in each year it was online.

DOI: https://doi.org/10.7554/eLife.45133.020

the highest publication rate of the bioRxiv categories: 51.5% of all bioRxiv evolutionary biology preprints have been published in a journal (*Figure 3b*). Examining the raw number of publications per category, neuroscience again comes out on top, with 2,608 preprints in that category published elsewhere (*Figure 3c*). When comparing the publication rates of preprints posted in each month we see that more recent preprints are published at a rate close to zero, followed by an increase in the rate of publication every month for about 12–18 months (*Figure 3a*). A similar dynamic was observed in a study of preprints posted to arXiv; after recording lower rates in the most recent time periods, Larivière et al. found publication rates of arXiv preprints leveled out at about 73% (*Larivière et al., 2014*). Of bioRxiv preprints posted between 2013 and the end of 2016, 67.0% have been published; if 2017 papers are included, that number falls to 64.0%. Of preprints posted in 2018, only 20.0% have been printed elsewhere (*Figure 3a*).

These publication statistics are based on data produced by bioRxiv's internal system that links publications to their preprint versions, a difficult challenge that appears to rely heavily on title-based matching. To better understand the reliability of the linking between preprints and their published versions, we selected a sample of 120 preprints that were not indicated as being published, and manually validated their publication status using Google and Google Scholar (see Methods). Overall, 37.5% of these 'unpublished' preprints had actually appeared in a journal. We found earlier years to have a much higher false-negative rate: 53% of the evaluated 'unpublished' preprints from 2015 had actually been published, though that number dropped to less than 17% in 2017 (*Figure 3—figure supplement 1*). While a more robust study would be required to draw more detailed conclusions about the

**Table 1.** Unique authors posting preprints in each year.

| Year | New authors | Total authors |
|---|---|---|
| 2013 | 608 | 608 |
| 2014 | 3,873 | 4,012 |
| 2015 | 7,584 | 8,411 |
| 2016 | 21,832 | 24,699 |
| 2017 | 52,051 | 61,239 |
| 2018 | 84,339 | 106,231 |

'New authors' counts authors posting preprints in that year that had never posted before; 'Total authors' includes researchers who may have already been counted in a previous year, but are also listed as an author on a preprint posted in that year. Data for table pulled directly from database. An SQL query to generate these numbers is provided in the Methods section.

DOI: https://doi.org/10.7554/eLife.45133.025

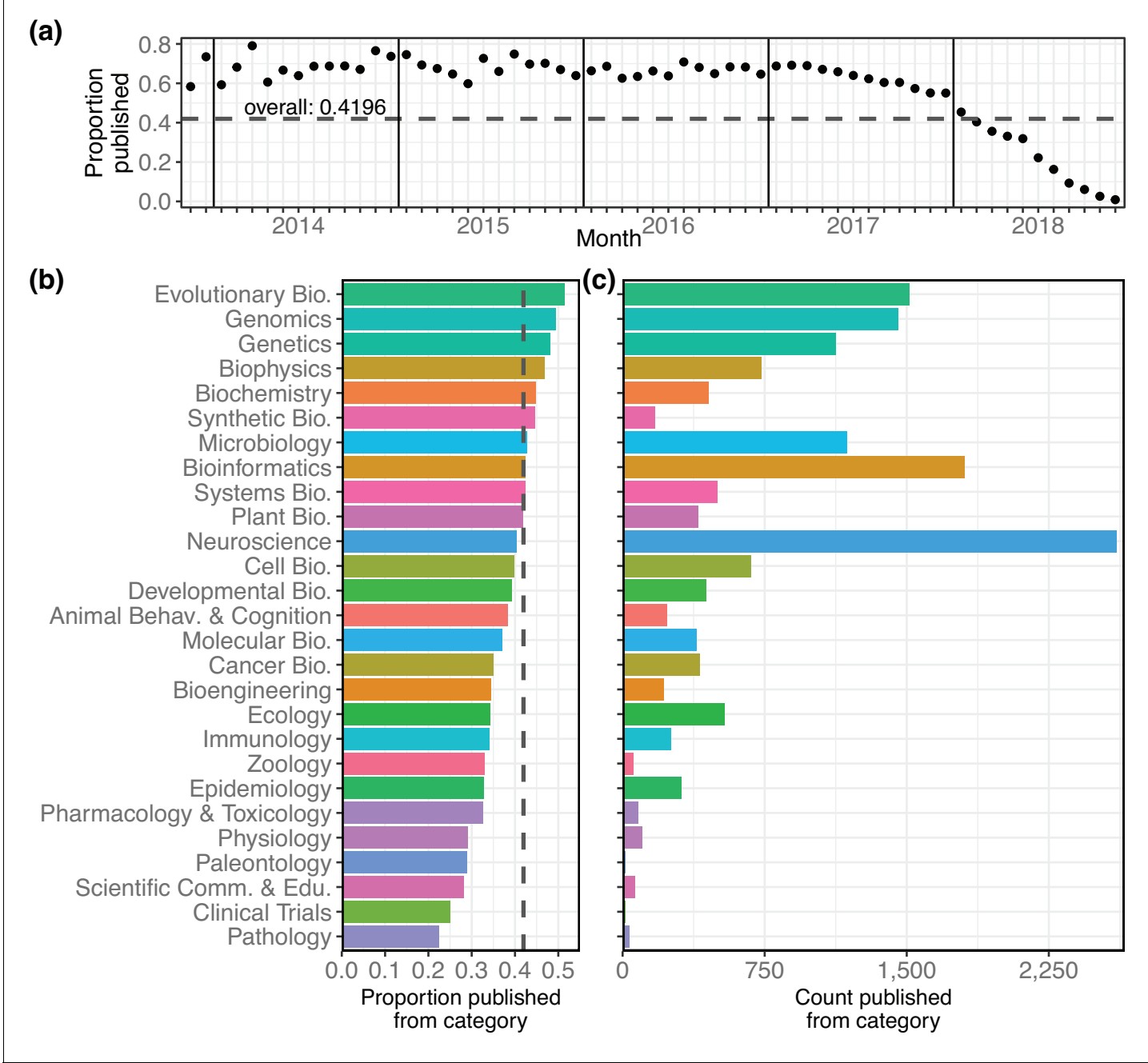

**Figure 3.** Characteristics of the bioRxiv preprints published in journals, across the 27 subject collections. (a) The proportion of preprints that have been published (y-axis), broken down by the month in which the preprint was first posted (x-axis). (b) The proportion of preprints in each category that have been published elsewhere. The dashed line marks the overall proportion of bioRxiv preprints that have been published and is at the same position as the dashed line in panel 3a. (c) The number of preprints in each category that have been published in a journal.

DOI: https://doi.org/10.7554/eLife.45133.026

The following source data and figure supplements are available for figure 3:

**Source data 1.** The number of preprints posted in each month, plus the count and proportion of those later published.
DOI: https://doi.org/10.7554/eLife.45133.029

**Source data 2.** The number of preprints posted in each category, plus the count and proportion of those published.
DOI: https://doi.org/10.7554/eLife.45133.030

**Figure supplement 1.** Observed annual publication rates and estimated range for actual publication rates.
DOI: https://doi.org/10.7554/eLife.45133.027

*Figure 3 continued on next page*

*Figure 3 continued*

**Figure supplement 1—source data 1.** Results of manual publication verification for a sample of bioRxiv preprints.
DOI: https://doi.org/10.7554/eLife.45133.028

'true' publication rate, this preliminary examination suggests the data from bioRxiv may be an underestimation of the number of preprints that have actually been published.

Overall, 15,797 bioRxiv preprints have appeared in 1,531 different journals (*Figure 4*). *Scientific Reports* has published the most, with 828 papers, followed by *eLife* and *PLOS ONE* with 750 and 741 papers, respectively. However, considering the proportion of preprints of the total papers published in each journal can lead to a different interpretation. For example, *Scientific Reports* published 398 bioRxiv preprints in 2018, but this represents 2.36% of the 16,899 articles it published in that year, as indexed by Web of Science (*Figure 4—source data 2*). In contrast, *eLife* published almost as many bioRxiv preprints (394), which means more than a third of their 1,172 articles from 2018 first appeared on bioRxiv. *GigaScience* had the highest proportion of articles from preprints in 2018 (49.4% of 89 articles), followed by *Genome Biology* (39.9% of 183 articles) and *Genome Research* (36.7% of 169 articles). Incorporating all years in which bioRxiv preprints have been published (2014–2018), these are also the three top journals.

Some journals have accepted a broad range of preprints, though none have hit all 27 of bioRxiv's categories – *PLOS ONE* has published the most diverse category list, with 26. (It has yet to publish a preprint from the clinical trials collection, bioRxiv's second-smallest.) Other journals are much more specialized, though in expected ways. Of the 172 bioRxiv preprints published by *The Journal of Neuroscience*, 169 were in neuroscience, and three were from animal behavior and cognition. Similarly, *NeuroImage* has published 211 neuroscience papers, two in bioinformatics, and one in bioengineering. It should be noted that these counts are based on the publications detected by bioRxiv and linked to their preprint, so some journals – for example, those that more frequently rewrite the titles of articles – may be underrepresented here.

When evaluating the downloads of preprints published in individual journals (*Figure 5*), there is a significant positive correlation between the median downloads per paper and journal impact factor (JIF): in general, journals with higher impact factors (*Clarivate Analytics, 2018*)

publish preprints that have more downloads. For example, *Nature Methods* (2017 JIF 26.919) has published 119 bioRxiv preprints; the median download count of these preprints is 2,266. By comparison, *PLOS ONE* (2017 JIF 2.766) has published 719 preprints with a median download count of 279 (*Figure 5*). In this analysis, each data point in the regression represented a journal, indicating its JIF and the median downloads per paper for the preprints it had published. We found a significant positive correlation between these two measurements (Kendall's $\tau_b$=0.5862, p=1.364e-06). We also found a similar, albeit weaker, correlation when we performed another analysis in which each data point represented a single preprint (n=7,445; Kendall's $\tau_b$=0.2053, p=9.311e-152; see Methods).

It is important to note that we did not evaluate when these downloads occurred, relative to a preprint's publication. While it looks like accruing more downloads makes it more likely that a preprint will appear in a higher impact journal, it is also possible that appearance in particular journals drives bioRxiv downloads after publication. The Rxivist dataset has already been used to begin evaluating questions like this (*Kramer, 2019*), and further study may be able to unravel the links, if any, between downloads and journals.

If journals are driving post-publication downloads on bioRxiv, however, their efforts are curiously consistent: preprints that have been published elsewhere have almost twice as many downloads as preprints that have not (*Table 2*; Mann–Whitney *U* test, p<2.2e-16). Among papers that have not been published, the median number of downloads per preprint is 208. For preprints that have been published, the median download count is 394 (Mann–Whitney *U* test, p<2.2e-16). When preprints published in 2018 are excluded from this calculation, the difference between published and unpublished preprints shrinks, but is still significant (*Table 2*; Mann–Whitney *U* test, p<2.2e-16). Though preprints posted in 2018 received more downloads in 2018 than preprints posted in previous years did (*Figure 2—figure supplement 3*), it appears they have not yet had time to accumulate as

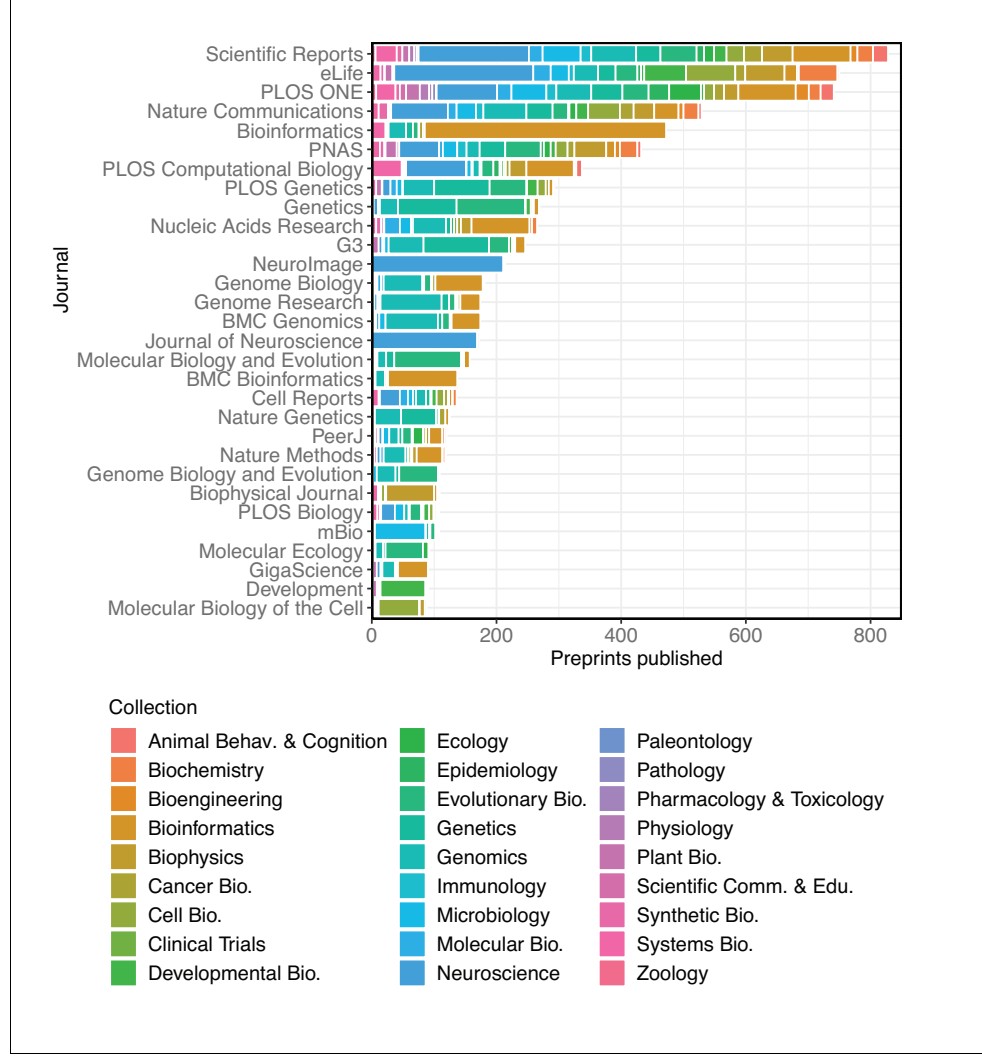

**Figure 4.** A stacked bar graph showing the 30 journals that have published the most bioRxiv preprints. The bars indicate the number of preprints published by each journal, broken down by the bioRxiv categories to which the preprints were originally posted.

DOI: https://doi.org/10.7554/eLife.45133.031

The following source data is available for figure 4:

**Source data 1.** The number of preprints published in each category by the 30 most prolific publishers of preprints.
DOI: https://doi.org/10.7554/eLife.45133.033

**Source data 2.** A table showing the proportion of published papers that were previously bioRxiv preprints, for the 30 journals that published the most bioRxiv preprints.
DOI: https://doi.org/10.7554/eLife.45133.034

**Source data 3.** Year-level data of the proportion of published papers that were previously bioRxiv preprints, for the 30 journals that published the most bioRxiv preprints.
DOI: https://doi.org/10.7554/eLife.45133.035

---

many downloads as papers from previous years (*Figure 2—figure supplement 4*).

We also retrieved the publication date for all published preprints using the Crossref 'Metadata Delivery' API (*Crossref, 2018*). This, combined with the bioRxiv data, gives us a comprehensive picture of the interval between the date a preprint is first posted to bioRxiv and the date it is published by a journal. These data show the median interval is 166 days, or about 5.5 months. 75% of preprints are published within 247 days of appearing on bioRxiv, and 90% are published within 346 days (*Figure 6a*). The median interval we found at the end of

November 2018 (166 days) is a 23.9% increase over the 134 day median interval reported by bioRxiv in mid-2016 (*Inglis and Sever, 2016*).

We also used these data to further examine patterns in the properties of the preprints that appear in individual journals. The journal that publishes preprints with the highest median age is *Nature Genetics*, whose median interval between bioRxiv posting and publication is 272 days (*Figure 6b*), a significant difference from every journal except *Genome Research* (Kruskal–

Wallis rank sum test, p<2.2e-16; Dunn's test q<0.05 comparing *Nature Genetics* to all other journals except *Genome Research*, after Benjamini–Hochberg correction). Among the 30 journals publishing the most bioRxiv preprints, the journal with the most rapid transition from bioRxiv to publication is *G3*, whose median, 119 days, is significantly different from all journals except *Genetics*, *mBio*, and *The Biophysical Journal* (*Figure 5*).

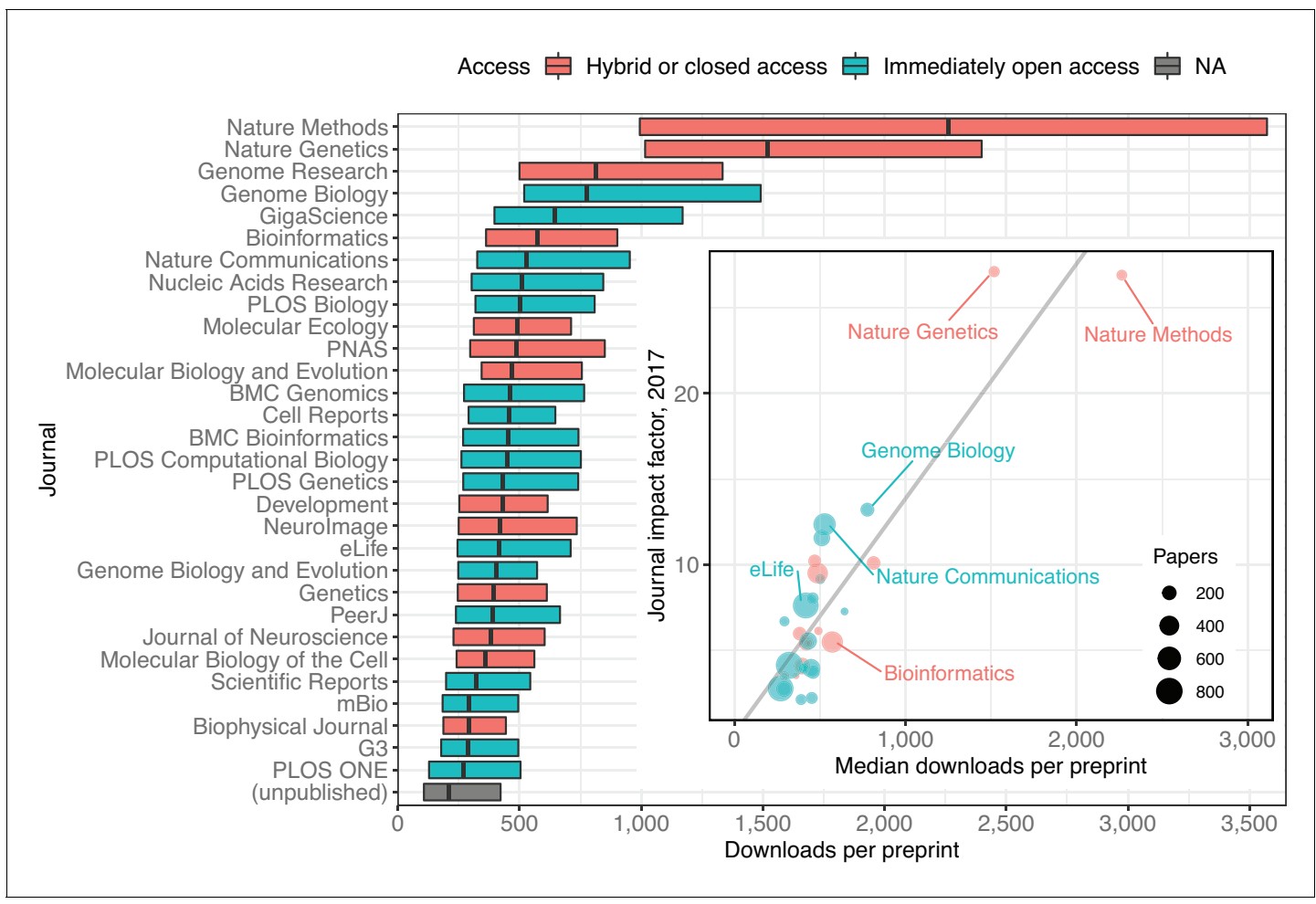

**Figure 5.** A modified box plot (without whiskers) illustrating the median downloads of all bioRxiv preprints published in a journal. Each box illustrates the journal's first quartile, median, and third quartile, as in *Figure 2c*. Colors correspond to journal access policy as described in the legend. Inset: A scatterplot in which each point represents an academic journal, showing the relationship between median downloads of the bioRxiv preprints published in the journal (x-axis) against its 2017 journal impact factor (y-axis). The size of each point is scaled to reflect the total number of bioRxiv preprints published by that journal. The regression line in this plot was calculated using the 'lm' function in the R 'stats' package, but all reported statistics use the Kendall rank correlation coefficient, which does not make as many assumptions about normality or homoscedasticity.
DOI: https://doi.org/10.7554/eLife.45133.036

The following source data is available for figure 5:

**Source data 1.** A list of every preprint with its total download count and the journal in which it was published, if any.
DOI: https://doi.org/10.7554/eLife.45133.037
**Source data 2.** Journal impact factor and access status of the 30 journals that have published the most preprints.
DOI: https://doi.org/10.7554/eLife.45133.038

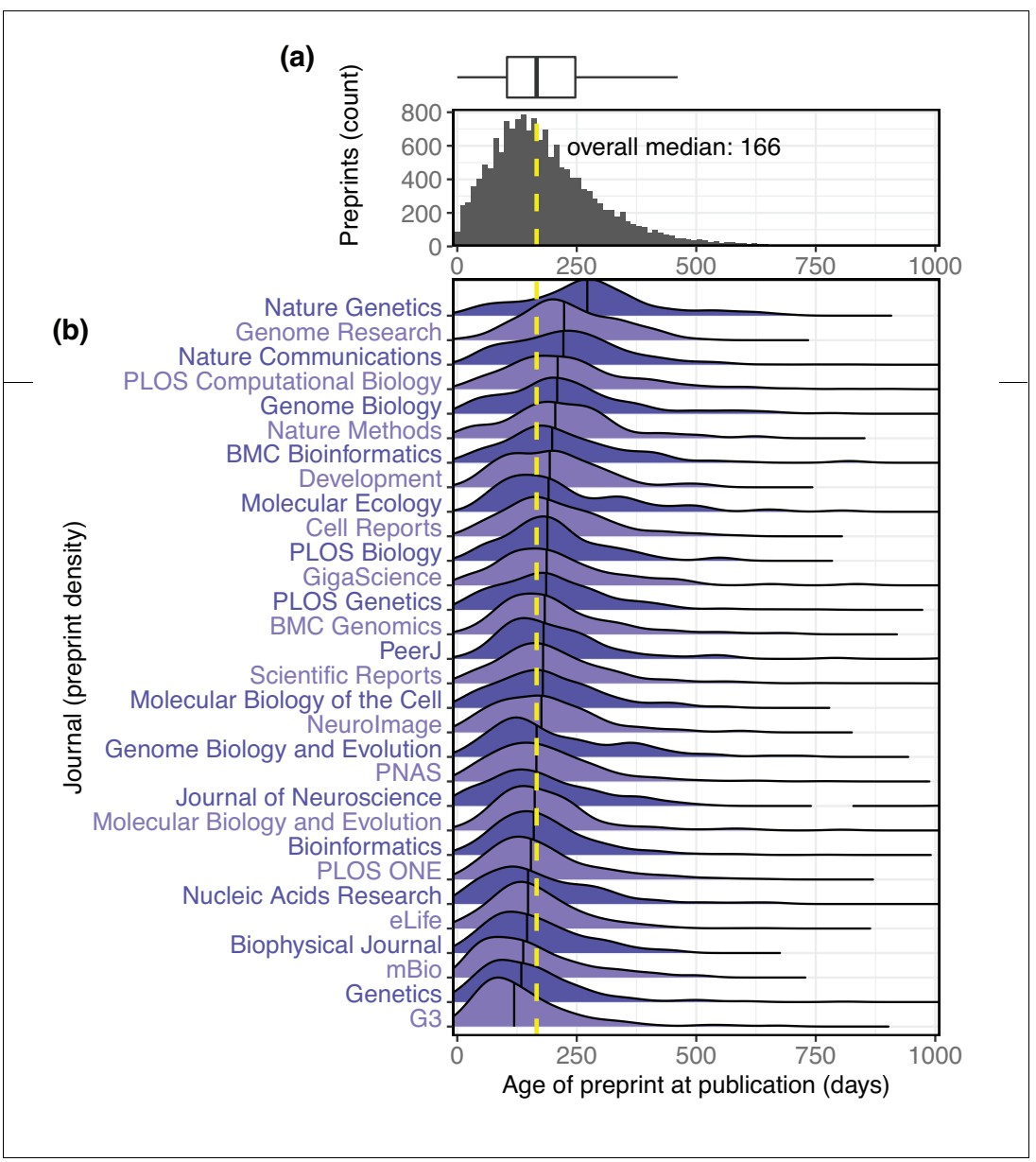

**Figure 6.** The interval between the date a preprint is posted to bioRxiv and the date it is first published elsewhere. (a) A histogram showing the distribution of publication intervals. The x-axis indicates the time between preprint posting and journal publication; the y-axis indicates how many preprints fall within the limits of each bin. The yellow line indicates the median; the same data is also visualized using a boxplot above the histogram. (b) The publication intervals of preprints, broken down by the journal in which each appeared. The journals in this list are the 30 journals that have published the most total bioRxiv preprints; the plot for each journal indicates the density distribution of the preprints published by that journal, excluding any papers that were posted to bioRxiv after publication. Portions of the distributions beyond 1,000 days are not displayed.

DOI: https://doi.org/10.7554/eLife.45133.041

The following source data is available for figure 6:

**Source data 1.** A list of every published preprint, the year it was first posted, the date it was published, and the interval between posting and publication, in days.

DOI: https://doi.org/10.7554/eLife.45133.042

**Source data 2.** A list of every preprint published in the 30 journals displayed in the figure, the journal in which it was published, and the interval between posting and publication, in days.

DOI: https://doi.org/10.7554/eLife.45133.043

*Figure 6 continued on next page*

*Figure 6 continued*

**Source data 3.** The results of Dunn's test, a pairwise comparison of the median publication interval of each journal in the figure.
DOI: https://doi.org/10.7554/eLife.45133.044

It is important to note that this metric does not directly evaluate the production processes at individual journals. Authors submit preprints to bioRxiv at different points in the publication process and may work with multiple journals before publication, so individual data points capture a variety of experiences. For example, 122 preprints were published within a week of being posted to bioRxiv, and the longest period between preprint and publication is 3 years, 7 months and 2 days, for a preprint that was posted in March 2015 and not published until October 2018 (*Figure 6a*).

## Discussion

Biology preprints have a growing presence in scientific communication, and we now have ongoing, detailed data to quantify this process. The ability to better characterize the preprint ecosystem can inform decision-making at multiple levels. For authors, particularly those looking for feedback from the community, our results show bioRxiv preprints are being downloaded more than one million times per month, and that an average paper can receive hundreds of downloads in its first few months online (*Figure 2— figure supplement 1*). *Serghiou and Ioannidis (2018)* evaluated download metrics for bioRxiv preprints through 2016 and found an almost identical median for downloads in a preprint's first month; we have expanded this to include more detailed longitudinal traffic metrics for the entire bioRxiv collection (*Figure 2b*).

For readers, we show that thousands of new preprints are being posted every month. This tracks closely with a widely referenced summary of submissions to preprint servers (*Pre-PubMed, 2018*) generated monthly by Pre-PubMed (http://www.prepubmed.org) and expands on submission data collected by researchers using custom web scrapers of their own (*Stuart, 2016*; *Stuart, 2017*; *Holdgraf, 2016*). There is also enough data to provide some evidence against the perception that research in preprint is less rigorous than papers appearing in journals (*Nature Biotechnology, 2017*; *Vale, 2015*). In short, the majority of bioRxiv preprints do appear in journals eventually, and potentially with very few differences: an

analysis of published preprints that had first been posted to arXiv.org found that 'the vast majority of final published papers are largely indistinguishable from their pre-print versions' (*Klein et al., 2016*). A 2016 project measured which journals had published the most bioRxiv preprints (*Schmid, 2016*); despite a six-fold increase in the number of published preprints since then, 23 of the top 30 journals found in their results are also in the top 30 journals we found (*Figure 5*).

For authors, we also have a clearer picture of the fate of preprints after they are shared online. Among preprints that are eventually published, we found that 75% have appeared in a journal by the time they had spent 247 days (about eight months) on bioRxiv. This interval is similar to results from Larivière et al. showing preprints on arXiv were most frequently published within a year of being posted there (*Larivière et al., 2014*), and to a later study examining bioRxiv preprints that found 'the probability of publication in the peer-reviewed literature was 48% within 12 months' (*Serghiou and Ioannidis, 2018*). Another study published in spring 2017 found that 33.6% of preprints from 2015 and earlier had been published (*Schloss, 2017*); our data through November 2018 show that 68.2% of preprints from 2015 and earlier have been published. Multiple studies have examined the interval between submission and publication at individual journals (e.g. *Himmelstein, 2016a*; *Royle, 2015*; *Powell, 2016*), but the incorporation of information about preprints is not as common.

We also found a positive correlation between the impact factor of journals and the number of downloads received by the preprints they have published. This finding in particular should be interpreted with caution. Journal impact factor is broadly intended to be a measurement of how citable a journal's 'average' paper is (*Garfield, 2006*), though it morphed long ago into an unfounded proxy for scientific quality in individual papers (*The PLoS Medicine Editors, 2006*). It is referenced here only as an observation about a journal-level metric correlated with preprint downloads; there is no indication that either factor is influencing the other, nor that

download numbers play a direct role in publication decisions.

More broadly, our granular data provide a new level of detail for researchers looking to evaluate many remaining questions. What factors may impact the interval between when a preprint is posted to bioRxiv and when it is published elsewhere? Does a paper's presence on bioRxiv have any relationship to its eventual citation count once it is published in a journal, as has been found with arXiv (e.g. *Feldman et al., 2018*; *Wang et al., 2018*; *Schwarz and Kennicutt, 2004*)? What can we learn from 'altmetrics' as they relate to preprints, and is there value in measuring a preprint's impact using methods rooted in online interactions rather than citation count (*Haustein, 2018*)? One study, published before bioRxiv launched, found a significant association between Twitter mentions of published papers and their citation counts (*Thelwall et al., 2013*) – have preprints changed this dynamic?

Researchers have used existing resources and custom scripts to answer questions like these. Himmelstein found that only 17.8% of bioRxiv papers had an 'open license' (*Himmelstein, 2016b*), for example, and another study examined the relationship between Facebook 'likes' of preprints and 'traditional impact indicators' such as citation count, but found no correlation for papers on bioRxiv (*Ringelhan et al., 2015*). Since most bioRxiv data is not programmatically accessible, many of these studies had to begin by scraping data from the bioRxiv website itself. The Rxivist API allows users to request the details of any preprint or author on bioRxiv, and the database snapshots enable bulk querying of preprints using SQL, C and several other languages (*PostgreSQL Global Development Group, 2018*) at a level of complexity currently unavailable using the standard bioRxiv web interface. Using these resources, researchers can now perform detailed and robust bibliometric analysis of the website with the largest collection of preprints in biology, the one that, beginning in September 2018, held more biology preprints than all other major preprint servers combined (*Anaya, 2018*).

In addition to our analysis here that focuses on big-picture trends related to bioRxiv, the Rxivist website provides many additional features that may interest preprint readers and authors. Its primary feature is sorting and filtering preprints based by download count or mentions on Twitter, to help users find preprints in particular categories that are being discussed either in the short term (Twitter) or over the span of months (downloads). Tracking these metrics could also help authors gauge public reaction to their work. While bioRxiv has compensated for a low rate of comments posted on the site itself (*Inglis and Sever, 2016*) by highlighting external sources such as tweets and blogs, Rxivist provides additional context for how a preprint compares to others on similar topics. Several other sites have attempted to use social interaction data to 'rank' preprints, though none incorporate bioRxiv download metrics. The 'Assert' web application (https://assert. pub) ranks preprints from multiple repositories based on data from Twitter and GitHub. The 'PromisingPreprints' Twitter bot (https://twitter. com/PromPreprint) accomplishes a similar goal, posting links to bioRxiv preprints that receive an exceptionally high social media attention score (*Altmetric Support, 2018*) from Altmetric (https://www.altmetric.com) in their first week on bioRxiv (*De Coster, 2017*). Arxiv Sanity Preserver (http://www.arxiv-sanity.com) provides rankings of arXiv.org preprints based on Twitter activity, though its implementation of this scoring (*Karpathy, 2018*) is more opinionated than that of Rxivist. Other websites perform similar curation, but based on user interactions within the sites themselves: SciRate (https://scirate. com), Paperkast (https://paperkast.com) and upvote.pub allow users to vote on articles that should receive more attention (*van der Silk et al., 2018*; *Özturan, 2018*), though upvote. pub is no longer online (*upvote.pub, 2018*). By comparison, Rxivist doesn't rely on user interaction – by pulling 'popularity' metrics from Twitter and bioRxiv, we aim to decouple the quality of our data from the popularity of the website itself.

In summary, our approach provides multiple perspectives on trends in biology preprints: (1) the Rxivist.org website, where readers can prioritize preprints and generate reading lists tailored to specific topics; (2) a dataset that can provide a foundation for developers and bibliometric researchers to build new tools, websites and studies that can further improve the ways we interact with preprints and (3) an analysis that brings together a comprehensive summary of trends in bioRxiv preprints and an examination of the crossover points between preprints and conventional publishing.

## Methods

### Data availability

There are multiple web links to resources related to this project:

- The Rxivist application is available on the web at https://rxivist.org and via Gopher at gopher://origin.rxivist.org.
- The source for the web crawler and API is available at https://github.com/blekhmanlab/rxivist (copy archived at https://github.com/elifesciences-publications/rxivist).
- The source for the Rxivist website is available at https://github.com/blekhmanlab/rxivist_web (copy archived at https://github.com/elifesciences-publications/rxivist_web).
- Data files used to generate the figures in this manuscript are available on Zenodo at https://doi.org/10.5281/zenodo.2465689, as is a snapshot of the database used to create the files.

### The Rxivist website

We attempted to put the Rxivist data to good use in a relatively straightforward web application. Its main offering is a ranked list of all bioRxiv preprints that can be filtered by areas of interest. The rankings are based on two available metrics: either the count of PDF downloads, as reported by bioRxiv, or the number of Twitter messages linking to that preprint, as reported by Crossref (https://crossref.org). Users can also specify a timeframe for the search – for example, one could request the most downloaded preprints in microbiology over the last two months, or view the preprints with the most Twitter activity since yesterday across all categories. Each preprint and each author is given a separate profile page, populated only by Rxivist data available from the API. These include rankings across multiple categories, plus a visualization of where the download totals for each preprint (and author) fall in the overall distribution across all 37,000 preprints and 170,000 authors.

### The Rxivist API and dataset

The full data described in this paper is available through Rxivist.org, a website developed for this purpose. BioRxiv data is available from Rxivist in two formats: (1) SQL 'database dumps' are currently pulled and published weekly on zenodo.org. (See Supplementary Information for a visualization and description of the schema.) These convert the entire Rxivist database into binary files that can be loaded by the free and open-source PostgreSQL database management system to provide a local copy of all collected data on every article and author on bioRxiv.org. (2) We also provide an API (application programming interface) from which users can request information in JSON format about individual preprints and authors, or search for preprints based on similar criteria available on the Rxivist website. Complete documentation is available at https://www.rxivist.org/docs.

While the analysis presented here deals mostly with overall trends on bioRxiv, the primary entity of the Rxivist API is the individual research preprint, for which we have a straightforward collection of metadata: title, abstract, DOI (digital object identifier), the name of any journal that has also published the preprint (and its new DOI), and which collection the preprint was submitted to. We also collected monthly traffic information for each preprint, as reported by bioRxiv. We use the PDF download statistics to generate rankings for each preprint, both site-wide and for each collection, over multiple timeframes (all-time, year to date, etc.). In the API and its underlying database schema, 'authors' exist separately from 'preprints' because an author can be associated with multiple preprints. They are recorded with three main pieces of data: name, institutional affiliation and a unique identifier issued by ORCID. Like preprints, authors are ranked based on the cumulative downloads of all their preprints, and separately based on downloads within individual bioRxiv collections. Emails are collected for each researcher, but are not necessarily unique (see 'Consolidation of author identities' below).

### Data acquisition

#### Web crawler design

To collect information on all bioRxiv preprints, we developed an application that pulled preprint data directly from the bioRxiv website. The primary issue with managing this data is keeping it up to date: Rxivist aims to essentially maintain an accurate copy of a subset of bioRxiv's production database, which means routinely running a web crawler against the website to find any new or updated content as it is posted. We have tried to find a balance between timely updates and observing courteous web crawler behavior; currently, each preprint is re-crawled once every two to three weeks to refresh its download metrics and publication status. The web crawler itself uses Python 3 and requires two primary modules for interacting with external services:

Requests-HTML (*Reitz, 2018*) is used for fetching individual web pages and pulling out the relevant data, and the psycopg2 module (*Di Gregorio and Varrazzo, 2018*) is used to communicate with the PostgreSQL database that stores all of the Rxivist data (*PostgreSQL Global Development Group, 2017*). PostgreSQL was selected over other similar database management systems because of its native support for text search, which, in our implementation, enables users to search for preprints based on the contents of their titles, abstracts and author list. The API, spider and web application are all hosted within separate Docker containers (*Docker Inc, 2018*), a decision we made to simplify the logistics required for others to deploy the components on their own: Docker is the only dependency, so most workstations and servers should be able to run any of the components.

New preprints are recorded by parsing the section of the bioRxiv website that lists all preprints in reverse-chronological order. At this point, a preprint's title, URL and DOI are recorded. The bioRxiv webpage for each preprint is then crawled to obtain details only available there: the abstract, the date the preprint was first posted, and monthly download statistics are pulled from here, as well as information about the preprint's authors – name, email address and institution. These authors are then compared against the list of those already indexed by Rxivist, and any unrecognized authors have profiles created in the database.

### Consolidation of author identities

Authors are most reliably identified across multiple papers using the bioRxiv feature that allows authors to specify an identifier provided by ORCID (https://orcid.org), a nonprofit that provides a voluntary system to create unique identification numbers for individuals. These ORCID (Open Researcher and Contributor ID) numbers are intended to serve approximately the same role for authors that DOIs do for papers (*Haak, 2012*), providing a way to identify individuals whose other information may change over time. 29,559 bioRxiv authors, or 17.4%, have an associated ORCID. If an individual included in a preprint's list of authors does not have an ORCID already recorded in the database, authors are consolidated if they have an identical name to an existing Rxivist author.

There are certainly authors who are duplicated within the Rxivist database, an issue arising mostly from the common complaint of

unreliable source data. 68.4% of indexed authors have at least one email address associated with them, for example, including 7,085 (4.40%) authors with more than one. However, of the 118,490 email addresses in the Rxivist database, 6,517 (5.50%) are duplicates that are associated with more than one author. Some of these are because real-life authors occasionally appear under multiple names, but other duplicates are caused by uploaders to bioRxiv using the same email address for multiple authors on the same preprint, making it far more difficult to use email addresses as unique identifiers. There are also cases like one from 2017, in which 16 of the 17 authors of a preprint were listed with the email address 'test@test.com.'

Inconsistent naming patterns cause another chronic issue that is harder to detect and account for. For example, at one point thousands of duplicate authors were indexed in the Rxivist database with various versions of the same name – including a full middle name, or a middle initial, or a middle initial with a period, and so on – which would all have been recorded as separate people if they did not all share an ORCID, to say nothing of authors who occasionally skip specifying a middle initial altogether. Accommodations could be made to account for inconsistencies such as these (using institutional affiliation or email address as clues, for example), but these methods also have the potential to increase the opposite problem of incorrectly combining different authors with similar names who intentionally introduce slight modifications such as a middle initial to help differentiate themselves. One allowance was made to normalize author names: when the web crawler searches for name matches in the database, periods are now ignored in string matches, so 'John Q. Public' would be a match with 'John Q Public.' The other naming problem we encountered was of the opposite variety: multiple authors with identical names (and no ORCID). For example, the Rxivist profile for author 'Wei Wang' is associated with 40 preprints and 21 different email addresses but is certainly the conglomeration of multiple researchers. A study of more than 30,000 Norwegian researchers found that when using full names rather than initials, the rate of name collisions was 1.4% (*Aksnes, 2008*).

### Retrieval of publication date information

Publication dates were pulled from the Crossref Metadata Delivery API (*Crossref, 2018*) using the publication DOI numbers provided by

bioRxiv. Dates were found for all but 31 (0.2%) of the 15,797 published bioRxiv preprints. Because journals measure publication date in different ways, several metrics were used. If a 'published—online' date was available from Crossref with a day, month and year, then that was recorded. If not, 'published—print' was used, and the Crossref 'created' date was the final option evaluated. Requests for which we received a 404 response were assigned a publication date of 1 Jan 1900, to prevent further attempts to fetch a date for those entries. It appears these articles were published, but with DOIs that were not registered correctly by the destination journal; for consistency, these results were filtered out of the analysis. There was no practical way to validate the nearly 16,000 values retrieved, but anecdotal evaluation reveals some inconsistencies. For example, the preprint with the longest interval before publication (1,371 days) has a publication date reported by Crossref of 1 Jul 2018, when it appeared in *IEEE/ACM Transactions on Computational Biology and Bioinformatics* 15(4). However, the IEEE website lists a date of 15 Dec 2015, two and a half years earlier, as that paper's 'publication date,' which they define as 'the very first instance of public dissemination of content.' Since every publisher is free to make their own unique distinctions, these data are difficult to compare at a granular level.

### Calculation of download rankings

The web crawler's 'ranking' step orders preprints and authors based on download count in two populations (overall and by bioRxiv category) and over several periods: all-time, year-to-date, and since the beginning of the previous month. The last metric was chosen over a 'month-to-date' ranking to avoid ordering papers based on the very limited traffic data available in the first days of each month – in addition to a short lag in the time bioRxiv takes to report downloads, an individual preprint's download metrics may only be updated in the Rxivist database once every two or three weeks, so metrics for a single month will be biased in favor of those that happen to have been crawled most recently. This effect is not eliminated in longer windows, but is diminished. The step recording the rankings takes a more unusual approach to loading the data. Because each article ranking step could require more than 37,000 'insert' or 'update' statements, and each author ranking requires more than 170,000 of the same, these modifications are instead written to a text

file on the application server and loaded by running an instance of the Postgres command-line client 'psql,' which can use the more efficient 'copy' command, a change that reduced the duration of the ranking process from several hours to less than one minute.

### Reporting of small p-values

In several locations, p-values are reported as <2.2e-16. It is important to note that this is an inequality, and these p-values are not necessarily identical. The upper limit, $2.2 \times 10^{-16}$, is not itself a particularly meaningful number and is an artifact of the limitations of the floating-point arithmetic used by R, the software used in the analysis. $2.2 \times 10^{-16}$ is the 'machine epsilon,' or the smallest number that can be added to 1.0 that would generate a result measurably different from 1.0. Though smaller numbers can be represented by the system, those smaller than the machine epsilon are not reported by default; we elected to do the same.

### *Data preparation*

Several steps were taken to organize the data that was used for this paper. First, the production data being used for the Rxivist API was copied to a separate 'schema' – a PostgreSQL term for a named set of tables. This was identical to the full database, but had a specifically circumscribed set of preprints. Once this was copied, the table containing the associations between authors and each of their papers ('article_authors') was pruned to remove references to any articles that were posted after 30 Nov 2018, and any articles that were not associated with a bioRxiv collection. For unknown reasons, 10 preprints (0.03%) could not be associated with a bioRxiv collection; because the bioRxiv profile page for some papers does not specify which collection it belongs to, these papers were ignored. Once these associations were removed, any articles meeting those criteria were removed from the 'articles' table. References to these articles were also removed from the table containing monthly bioRxiv download metrics for each paper ('article_traffic'). We also removed all entries from the 'article_traffic' table that recorded downloads after November 2018. Next, the table containing author email addresses ('author_emails') was pruned to remove emails associated with any author that had zero preprints in the new set of papers; those authors were then removed from the 'authors' table.

Before evaluating data from the table linking published preprints to journals and their post-publication DOI ('article_publications'), journal names were consolidated to avoid under-counting journals with spelling inconsistencies. First, capitalization was stripped from all journal titles, and inconsistent articles ('The Journal of…' vs. 'Journal of…'; 'and' vs. '&' and so on) were removed. Then, the list of journals was reviewed by hand to remove duplication more difficult to capture automatically: 'PNAS' and 'Proceedings of the National Academy of Sciences,' for example. Misspellings were rare, but one publication in 'integrrative biology' did appear. See *figures. md* in the project's GitHub repository ([https:// github.com/blekhmanlab/rxivist/blob/master/ paper/figures.md](https://github.com/blekhmanlab/rxivist/blob/master/paper/figures.md)) for a full list of corrections made to journal titles. We also evaluated preprints for publication in 'predatory journals,' organizations that use irresponsibly low academic standards to bolster income from publication fees (*Xia et al., 2015*). A search for 1,345 journals based on the list compiled by Stop Predatory Journals ([https://predatoryjournals. com](https://predatoryjournals.com)) showed that bioRxiv publication data did not include any instances of papers appearing in those journals (*Stop Predatory Journals, 2018*). It is important to note that the absence of this information does not necessarily indicate that preprints have not appeared in these journals – we performed this search to ensure our analysis of publication rates was not inflated with numbers from illegitimate publications.

### *Data analysis*
Reproduction of figures
Two files are needed to recreate the figures in this manuscript: a compressed database backup containing a snapshot of the data used in this analysis, and a file called *figures.md* storing the SQL queries and R code necessary to organize the data and draw the figures. The PostgreSQL documentation for restoring database dumps should provide the necessary steps to 'inflate' the database snapshot, and each figure and table is listed in *figures.md* with the queries to generate comma-separated values files that provide the data underlying each figure. (Those who wish to skip the database reconstruction step will find CSVs for each figure provided along with these other files.) Once the data for each figure is pulled into files, executing the accompanying R code should create figures containing the exact data as displayed here.

Tallying institutional authors and preprints
When reporting the counts of bioRxiv authors associated with individual universities, there are several important caveats. First, these counts only include the most recently observed institution for an author on bioRxiv: if someone submits 15 preprints at Stanford, then moves to the University of Iowa and posts another preprint afterward, that author will be associated with the University of Iowa, which will receive all 16 preprints in the inventory. Second, this count is also confounded by inconsistencies in the way authors report their affiliations: for example, 'Northwestern University,' which has 396 preprints, is counted separately from 'Northwestern University Feinberg School of Medicine,' which has 76. Overlaps such as these were not filtered, though commas in institution names were omitted when grouping preprints together.

Evaluation of publication rates
Data referenced in this manuscript is limited to preprints posted through the end of November 2018. However, determining which preprints had been published in journals by the end of November required refreshing the entries for all 37,000 preprints after the month ended. Consequently, it is possible that papers published after the end of November (but not after the first weeks of December) are included in the publication statistics.

Estimation of ranges for true publication rates
To evaluate the sensitivity of the system bioRxiv uses to detect published versions of preprints, we pulled a random sample of 120 preprints that had not been marked as published on bioRxiv.org – 30 preprints from each year between 2014 and 2017. We then performed a manual online literature search for each paper to determine whether they had been published. The primary search method was searching on Google. com for the preprint's title and the senior author's last name. If this did not return any results that looked like publications, other author names were added to the search to replace the senior author's name. If this did not return any positive results, we also checked Google Scholar ([https://scholar.google.com](https://scholar.google.com)) for papers with similar titles. If any of the preprint's authors, particularly the first and last authors, had Google Scholar profiles, they were reviewed for publications on subject matter similar to the preprint. If a publication looked similar to the preprint, a visual comparison between the

preprint and published paper's abstract and introduction was used to determine if they were simply different versions of the same paper. The paper was marked as a true negative if none of these returned positive results, or if the suspected published paper described a study that was different enough that the preprint effectively described a different research project.

Once all 120 preprints had been evaluated, the results were used to approximate a false-negative rate to each year – the proportion of preprints that had been incorrectly excluded from the list of published papers. The sample size for each year (30) was used to calculate the margin of error using a 95% confidence interval (17.89 percentage points). This margin was then used to generate the minimum and maximum false-negative rates for each year, which were then used to calculate the minimum and maximum number of incorrectly classified preprints from each year. These numbers yielded a range for each year's actual publication rate; for 2015, for example, bioRxiv identified 1,218 preprints (out of 1,774) that had been published. The false-negative rate and margin of error suggest between 197 and 396 additional preprints have been published but not detected, yielding a final range of 1,415–1,614 preprints published in that year.

To evaluate the specificity of the publication detection system, we pulled 40 samples (10 from each of the years listed above) that bioRxiv had listed as published, and found that all 40 had been accurately classified. Though this helps establish that bioRxiv is not consistently finding all preprint publications, it should be noted that the determination of a more precise estimation for publication rates would require deeper analysis and sampling.

### Calculation of publication intervals

There are 15,797 distinct preprints with an associated date of publication in a journal, a corpus too large to allow detailed manual validation across hundreds of journal websites. Consequently, these dates are only as accurate as the data collected by Crossref from the publishers. We attempted to use the earliest publication date, but researchers have found that some publishers may be intentionally manipulating dates associated with publication timelines (*Royle, 2015*), particularly the gap between online and print publication, which can inflate journal impact factor (*Tort et al., 2012*). Intentional or not, these gaps may be inflating the time to press measurements of some preprints

and journals in our analysis. In addition, there are 66 preprints (0.42%) that have a publication date that falls before the date it was posted to bioRxiv; these were excluded from analyses of publication interval.

### Counting authors with middle initials

To obtain the comparatively large counts of authors using one or two middle initials, results from a SQL query were used without any curation. For the counts of authors with three or four middle initials, the results of the database call were reviewed by hand to remove 'author' names that look like initials, but are actually the name of consortia ('International IBD Genetics Consortium') or authors who provided non-initialized names using all capital letters.

### Acknowledgements

We thank the members of the Blekhman lab, Kevin M Hemer, and Kevin LaCherra for helpful discussions. We also thank the bioRxiv staff at Cold Spring Harbor Laboratory for building a valuable tool for scientific communication, and also for not blocking our web crawler even when it was trying to read every web page they have. We are grateful to Crossref for maintaining an extensive, freely available database of publication data.

**Richard J Abdill** is in the Department of Genetics, Cell Biology, and Development, University of Minnesota, Minneapolis, United States
https://orcid.org/0000-0001-9565-5832

**Ran Blekhman** is in the Department of Genetics, Cell Biology, and Development, and the Department of Ecology, Evolution, and Behavior, University of Minnesota, Minneapolis, United States
blekhman@umn.edu
http://orcid.org/0000-0003-3218-613X

*Author contributions:* Richard J Abdill, Data curation, Software, Formal analysis, Visualization, Writing—original draft; Ran Blekhman, Conceptualization, Methodology, Writing—review and editing

*Competing interests:* The authors declare that no competing interests exist.

## Funding

| Funder | Grant reference number | Author |
|---|---|---|
| College of Biological Sciences, University of Minnesota | | Ran Blekhman |
| National Institutes of Health | R35-GM128716 | Ran Blekhman |
| University of Minnesota | McKnight Land-Grant Professorship | Ran Blekhman |

The funders had no role in study design, data collection and interpretation, or the decision to submit the work for publication.

## Decision letter and Author response

Decision letter https://doi.org/10.7554/eLife.45133.053
Author response https://doi.org/10.7554/eLife.45133.054

## Additional files

### Supplementary files
• Source code 1. SQL queries and R code required to pull the data and visualize each figure.
DOI: https://doi.org/10.7554/eLife.45133.045

• Supplementary file 1. Detailed description of all database fields and tables.
DOI: https://doi.org/10.7554/eLife.45133.046

• Transparent reporting form
DOI: https://doi.org/10.7554/eLife.45133.047

### Data availability

Source data for all figures have been provided in supporting files. A database snapshot containing all data collected for this study has been deposited in a Zenodo repository with DOI 10.5281/zenodo.2465688.

The following datasets were generated:

| Author(s) | Year | Dataset URL | Database and Identifier |
|---|---|---|---|
| Abdill RJ, Blekhman R | 2019 | http://doi.org/10.5281/zenodo.2465688 | Zenodo, 10.5281/zenodo.2465688 |
| Abdill RJ, Blekhman R | 2019 | http://doi.org/10.5281/zenodo.2529922 | Zenodo, 10.5281/zenodo.2529922 |

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
