## [Decision Letter]

Thank you for submitting your article "Tracking the popularity and outcomes of all bioRxiv preprints" for consideration by *eLife*. Your article has been reviewed by three peer reviewers, and the evaluation has been overseen by Emma Pewsey (Associate Features Editor) and Peter Rodgers (Features Editor). The following individual involved in review of your submission has agreed to reveal his identity: Casey S Greene (Reviewer #2). The other two reviewers remain anonymous.

The reviewers have discussed the reviews with one another and the Associate Features Editor has drafted this decision to help you prepare a revised submission. In addition, a member of Reviewer #2's lab reviewed the manuscript for the life sciences overlay biOverlay (https://www.bioverlay.org/post/2019-03-tracking-the-popularity-and-outcomes-of-all-biorxiv-preprints/). The most important comments in that review are included in the points below. If there are additional comments from biOverlay that you wish to address in the revision please highlight these in your author response.

Summary:

This is the most comprehensive analysis to date of preprints in the life sciences. Although similar analysis of the preprint ecosystem has occurred with respect to the preprint server arXiv, which serves the physics, mathematics and computer science communities, similar analysis has not been done with preprint servers that post life sciences research. Therefore, this study is both welcome and important, providing very useful information for those considering whether to post preprints and those wondering how to properly evaluate them. The reviewers also praise the web application developed by the authors that allows users to access preprint metadata, which is of broad interest in the community.

Essential revisions:

1) The analysis of publications outcomes provides a lower bound to the proportion of preprints that are eventually published. The bioRxiv-journal linking is somewhat noisy. For example, Reviewer #2 states: "On some occasions I have had to send an email to bioRxiv to establish the link for my own preprints, and have also found that other 'unpublished' preprints have occasionally already been published." To check the veracity of the reported numbers, the authors should randomly select ~100 manuscripts (perhaps stratified by year for 2014, 2015, 2016, and 2017) that are in the not-yet-published category and use Google and Google Scholar to measure the fraction that are published but missed by the bioRxiv process.

2) The authors interpret downloads as the number of people reading the paper: "We find preprints are being read more than ever before (1.1 million downloads in October 2018 alone)." These are not strictly the same and the authors should use downloads as the correct metric in the Abstract. In their Results, they should then clarify that they use downloads as a metric for readers. Instead of "Considering the number of downloads for each preprint, we find that bioRxiv's usage among readers is also increasing rapidly:" they could write: "Using downloads as a metric for readers' usage of preprints, we find that bioRxiv's usage among readers is also increasing rapidly."

3) To maximize the value of the resource and to reduce confusion around re-use attempts, the authors should apply the CC0 license to the work. The justification from Daniel Himmelstein at biOverlay provides a good rationale as to why this is likely to be the right choice:

"The Zenodo archive with the data for the study and the Zenodo archive with the current database are currently released under a CC BY-NC 4.0 License. This license forbids commercial reuse and hence is not considered an open license. Furthermore, it is a poor choice for data. First, it is unclear whether copyright applies to any aspects of the created database in the United States. Therefore, some users may decide that either no copyright applies to the data or that their reuse is fair use. For these users, the CC license is irrelevant and can be ignored. However, more cautious users or those in other jurisdictions may feel restricted by the license and hence not use the data. The NC stipulation makes the data difficult to integrate with other data. For example, if copyright does apply, then the data would be incompatible with data licensed under CC BY-SA (share alike). Finally, attribution on a dataset level is often challenging, and social norms rather than legal recourse are generally sufficient. The authors should look into placing their datasets into the public domain via a CC0 waiver/license, which is becoming common and enables the information to be ingested by other data commons such as Wikidata. Finally, it is possible that users could rerun the open source code to regenerate the database, thereby creating a parallel version that is unencumbered by potential copyright issues."

4) While the correlation between download and journal impact factor may be informative, using this metric is also problematic to assess a paper's true impact. The Discussion should address this issue.

5) In the Discussion, the authors write that "the Rxivist website provides many additional features that may interest preprint readers." The authors should also mention that this web application may interest preprint authors, as a method to assess community interest in their work. This could be especially important given the low rate of comments on preprints posted on bioRxiv (Inglis and Sever, 2016).

Minor points:

6) The authors show that published preprints are downloaded more often than unpublished preprints: "Site-wide, the median number of downloads per preprint is 208, among papers that have not been published. For preprints that have been published, the median download count is 394 (Mann-Whitney U test, p < 2.2e-16). When preprints published in 2018 are excluded from this calculation, the difference between published and unpublished preprints shrinks, but is still significant (Table 2; Mann-Whitney U test, p < 2.2e-16)." Despite the difference between published and unpublished shrinking, the p value is similar to that of the data that includes preprints posted in 2018. Is this accurate?

7) Please clarify what the dashed lines represent in Figure 2–figure supplement 1 and Figure 2–figure supplement 4.

Optional suggestions:

8) Figure 1 shows the number of preprints per field; however, different fields produce different numbers of papers. It would be helpful to provide an estimate of the relative size of each field to help understand the proportion of papers that are also submitted as preprints. The ideal analysis would include the number of bioRxiv postings by subject category vs. the number of papers that appear on PubMed for the subject category. A commenter on biOverlay mentions that they may be able to assist with this analysis: see https://hyp.is/AchpIjxgEemwB89_ndsvaw/www.bioverlay.org/post/2019-03-tracking-the-popularity-and-outcomes-of-all-biorxiv-preprints/

9) The question of whether or not paywalled articles get more preprint downloads following journal publication came up in our journal club. It may be interesting to mention the existing work to date on this topic in the discussion section: see the following hypothes.is comment on biOverlay: https://hyp.is/ThpfijxeEemiId-NPKT3nA/www.bioverlay.org/post/2019-03-tracking-the-popularity-and-outcomes-of-all-biorxiv-preprints/

10) An automatically generated visualization of the database schema in the methods would be helpful to reusers.

---

## [Author Response]

We are grateful to the reviewers for their thoughtful and detailed comments, and we have made changes to address the concerns raised both in the official reviews and in the biOverlay assessment that was referenced in your March 5 decision letter.

Our revised manuscript, which we believe is much improved, includes several edits detailed below in a point-by-point response to the reviewers’ comments. Major changes include augmenting three sections with additional analysis:

1) A new analysis measuring preprint publications per journal as a proportion of their total articles (new Figure 4–figure supplement 1 and new paragraph in the Results section).

2) A new evaluation of the accuracy of bioRxiv data on the publication of preprints (new Figure 3–figure supplement 1, new paragraph in the Results section, and new subsection in the Methods).

3) A more detailed regression analysis for the relationship between Journal Impact Factor and the downloads of preprints (new text in the Results section) and a clarification on the use of Journal Impact Factor in our analysis (new paragraph in the Discussion section).

In addition, we have added several new points to the Discussion section, and have clarified wording and terminology where requested.

Essential revisions:1) The analysis of publications outcomes provides a lower bound to the proportion of preprints that are eventually published. The bioRxiv-journal linking is somewhat noisy. For example, Reviewer #2 states: "On some occasions I have had to send an email to bioRxiv to establish the link for my own preprints, and have also found that other 'unpublished' preprints have occasionally already been published." To check the veracity of the reported numbers, the authors should randomly select ~100 manuscripts (perhaps stratified by year for 2014, 2015, 2016, and 2017) that are in the not-yet-published category and use Google and Google Scholar to measure the fraction that are published but missed by the bioRxiv process.

We thank the reviewers for this important suggestion, which we believe provides valuable context to the "Publication outcomes" section of the manuscript. To better understand the reliability of the linking between preprints and their published versions, we selected 30 preprints for each year between 2014 and 2017 that were not indicated as being published, and manually validated their publication status using Google and Google Scholar. Overall, 37.5% of the 120 “unpublished” preprints that we evaluated had actually been published. The following description of these findings was added to the Results section:

“These publication statistics are based on data produced by bioRxiv’s internal system that links publications to their preprint versions, a difficult challenge that appears to rely heavily on title-based matching. To better understand the reliability of the linking between preprints and their published versions, we selected a sample of 120 preprints that were not indicated as being published, and manually validated their publication status using Google and Google Scholar (see Methods). Overall, 37.5% of these “unpublished” preprints had actually appeared in a journal. We found earlier years to have a much higher false-negative rate: 53 percent of the evaluated "unpublished" preprints from 2015 had actually been published, though that number dropped to less than 17 percent in 2017 (Figure 3–figure supplement 1). While a more robust study would be required to draw more detailed conclusions about the “true” publication rate, this preliminary examination suggests the data from bioRxiv may be an underestimation of the number of preprints that have actually been published.”

These results are summarized in a new supplementary figure (Figure 3–figure supplement 1), illustrating the observed publication rate in bioRxiv data (dots) and the range (error bars) of the actual publication rate, as suggested by the samples.

The ranges were calculated by determining the false-negative rate for each year (published papers not listed as published on bioRxiv), and using the margin of error for our survey (~17%) to find the minimum and maximum false-negative rate at a 95% confidence interval. These percentages were then applied to the number of "unpublished" papers in each year and added to the "published" count provided by bioRxiv.

In addition to the explanation that observed publication rates are likely an underestimation, we added another caveat to the Results section describing the journals that have published the most preprints, to clarify that some journals or fields may have conventions that impair automatic linking:

“Some journals have accepted a broad range of preprints, though none have hit all 27 of bioRxiv’s categories – PLOS ONE has published the most diverse category list, with 26. (It has yet to publish a preprint from the clinical trials collection, bioRxiv's second-smallest.) Other journals are much more specialized, though in expected ways: Of the 172 bioRxiv preprints published by The Journal of Neuroscience, 169 were in neuroscience, and 3 were from animal behavior and cognition. Similarly, NeuroImage has published 211 neuroscience papers, 2 in bioinformatics, and 1 in bioengineering. *It should be noted that these counts are based on the publications detected by bioRxiv and linked to their preprint, so some journals – for example, those that more frequently rewrite the titles of articles – may be underrepresented here.*”

We added a subsection to the Materials and methods section explaining the details of the analysis: see section **“**Estimation of ranges for true publication rates.”

2) The authors interpret downloads as the number of people reading the paper: "We find preprints are being read more than ever before (1.1 million downloads in October 2018 alone)." These are not strictly the same and the authors should use downloads as the correct metric in the Abstract. In their Results, they should then clarify that they use downloads as a metric for readers. Instead of "Considering the number of downloads for each preprint, we find that bioRxiv's usage among readers is also increasing rapidly:" they could write: "Using downloads as a metric for readers' usage of preprints, we find that bioRxiv's usage among readers is also increasing rapidly."

We agree with the reviewers’ point that PDF download counts do not necessarily represent a one-to-one indication of readership. To clarify that this proxy was being used, we revised the sentence in question in the Results section:

“Using preprint downloads as a metric for readership, we find that bioRxiv’s usage among readers is also increasing rapidly (Figure 2). The total download count in October 2018 (1,140,296) was an 82 percent increase over October 2017, which itself was a 115 percent increase over October 2016 (Figure 2A).”

We have also changed the referenced sentence in the Abstract to remove the inference between a paper being downloaded and a paper being read:

“We find preprints are being downloaded more than ever before (1.1 million tallied in October 2018 alone) and that the rate of preprints being posted has increased to a recent high of 2,100 per month.”

3) To maximize the value of the resource and to reduce confusion around re-use attempts, the authors should apply the CC0 license to the work. The justification from Daniel Himmelstein at biOverlay provides a good rationale as to why this is likely to be the right choice:"The Zenodo archive with the data for the study and the Zenodo archive with the current database are currently released under a CC BY-NC 4.0 License. This license forbids commercial reuse and hence is not considered an open license. Furthermore, it is a poor choice for data. First, it is unclear whether copyright applies to any aspects of the created database in the United States. Therefore, some users may decide that either no copyright applies to the data or that their reuse is fair use. For these users, the CC license is irrelevant and can be ignored. However, more cautious users or those in other jurisdictions may feel restricted by the license and hence not use the data. The NC stipulation makes the data difficult to integrate with other data. For example, if copyright does apply, then the data would be incompatible with data licensed under CC BY-SA (share alike). Finally, attribution on a dataset level is often challenging, and social norms rather than legal recourse are generally sufficient. The authors should look into placing their datasets into the public domain via a CC0 waiver/license, which is becoming common and enables the information to be ingested by other data commons such as Wikidata. Finally, it is possible that users could rerun the open source code to regenerate the database, thereby creating a parallel version that is unencumbered by potential copyright issues."

We fully agree that open licensing is important, both for transparency and to make it easier for future researchers to use the information. The repository holding the raw data from our manuscript (https://doi.org/10.5281/zenodo.2603083) has been switched to the CC0 license.

4) While the correlation between download and journal impact factor may be informative, using this metric is also problematic to assess a paper's true impact. The Discussion should address this issue.

We agree that JIF is a poor measure of impact for an individual article—indeed, JIF is a problematic metric even for what it claims to represent. We have added a paragraph to the Discussion section that clarifies the restricted applicability of this finding:

“We also found a positive correlation between the impact factor of journals and the number of downloads received by the preprints they have published. This finding in particular should be interpreted with caution. Journal impact factor is broadly intended to be a measurement of how citable a journal’s “average” paper is (Garfield 2006), though it morphed long ago into an unfounded proxy for scientific quality in individual papers (“The Impact Factor Game” 2006). It is referenced here only as an observation about a *journal*-level metric correlated with preprint downloads: There is no indication that either factor is influencing the other, nor that download numbers play a direct role in publication decisions.”

We also received feedback from the community about the use of journal impact factor in this analysis, including questions regarding the strength of the correlation observed between preprint downloads and JIF. The initial analysis, of the 30 journals that had published the most bioRxiv preprints, revealed a correlation between a journal's JIF and the median download count of the preprints it published. To strengthen this evaluation, we performed another regression analysis in which the data points were each individual preprint published by those journals, referenced against the impact factor of the journal in which it appeared. Both analyses found positive correlations between downloads and JIF, and are explained in the Results section:

“In our analysis, each data point in the regression represented a journal, indicating its JIF and the median downloads per paper for the preprints it had published. We found a significant positive correlation between these two measurements (Kendall’s τ_b_=0.5862, p=1.364e-06). We also found a similar, albeit weaker, correlation when we performed another analysis in which each data point represented a single preprint (n=7,445; Kendall’s τ_b_=0.2053, p=9.311e-152; see Materials and methods).”

5) In the Discussion, the authors write that "the Rxivist website provides many additional features that may interest preprint readers." The authors should also mention that this web application may interest preprint authors, as a method to assess community interest in their work. This could be especially important given the low rate of comments on preprints posted on bioRxiv (Inglis and Sever, 2016).

We have revised the sentence in question to make it clearer that the Rxivist website may have features that are helpful not just to readers but also to preprint authors looking to place their preprint's performance in context. The paragraph now reads:

“In addition to our analysis here focused on big-picture trends related to bioRxiv, the Rxivist website provides many additional features that may interest preprint readers and authors. Its primary feature is sorting and filtering preprints based by download count or mentions on Twitter, to help users find preprints in particular categories that are being discussed either in the short term (Twitter) or over the span of months (downloads). Tracking these metrics could also help authors gauge public reaction to their work: While bioRxiv has compensated for a low rate of comments posted on the site itself (Inglis and Sever, 2016) by highlighting external sources such as tweets and blogs, Rxivist provides additional context for how a preprint compares to others on similar topics.”

Minor points:6) The authors show that published preprints are downloaded more often than unpublished preprints: "Site-wide, the median number of downloads per preprint is 208, among papers that have not been published. For preprints that have been published, the median download count is 394 (Mann-Whitney U test, p < 2.2e-16). When preprints published in 2018 are excluded from this calculation, the difference between published and unpublished preprints shrinks, but is still significant (Table 2; Mann-Whitney U test, p < 2.2e-16)." Despite the difference between published and unpublished shrinking, the p value is similar to that of the data that includes preprints posted in 2018. Is this accurate?

We appreciate the reviewers' attention to detail in what could have been an unfortunate error. It is important to clarify that the p-values are not necessarily similar in this case: we state that these p-values are lower than 2.2e-16, but not that they are otherwise similar. Both inequalities in question (and another elsewhere in the text) are reported that way because of the limitations of the floating point arithmetic used by R, which is the software used in our analyses. We have added a paragraph to the Methods section to clarify this: see section “Reporting of small p-values”.

7) Please clarify what the dashed lines represent in Figure 2–figure supplement 1 and Figure 2–figure supplement 4.

A sentence has been added to the end of the legend for Figure 2–figure supplement 1 to clarify that the dashed lines indicate medians: “The dashed line represents the median downloads per month over each paper’s first 12 months.”

Optional suggestions:8) Figure 1 shows the number of preprints per field; however, different fields produce different numbers of papers. It would be helpful to provide an estimate of the relative size of each field to help understand the proportion of papers that are also submitted as preprints. The ideal analysis would include the number of bioRxiv postings by subject category vs. the number of papers that appear on PubMed for the subject category. A commenter on biOverlay mentions that they may be able to assist with this analysis: see https://hyp.is/AchpIjxgEemwB89_ndsvaw/www.bioverlay.org/post/2019-03-tracking-the-popularity-and-outcomes-of-all-biorxiv-preprints/

We agree that the analysis of absolute numbers of preprints in each bioRxiv category would benefit from field-specific context. However, it would be a large undertaking, as this would require evaluating every article in PubMed and determining the bioRxiv category to which each would theoretically have been posted. We attempted to use bioRxiv abstracts to train a Random Forest classifier to assign putative categories, but validation using other bioRxiv abstracts suggested that our initial attempt did not pull enough distinctive information from the abstracts to make confident predictions about categorization. However, the linked comment appears to be referring to a slightly different analysis recommended by Dr Himmelstein, which we have added to our submission: the proportion of papers from each journal that first appeared on bioRxiv. While this does not measure field-specific enthusiasm for preprints, using individual journals as samples for their fields of focus does provide some insight into the prevalence of preprints. We found that journals that published large numbers of preprints were not necessarily "enthusiastic" about preprints in particular—for example, Scientific Reports has published more bioRxiv preprints than any other journal, but these represent a tiny fraction of their published articles. Others, such as Genome Biology and GigaScience, publish fewer articles that are more consistently "preprinted" first. This information has been included as a table labelled Figure 4–source data 2. We have also added this information to the section of the results that describes Figure 4.

9) The question of whether or not paywalled articles get more preprint downloads following journal publication came up in our journal club. It may be interesting to mention the existing work to date on this topic in the discussion section: see the following hypothes.is comment on biOverlay: https://hyp.is/ThpfijxeEemiId-NPKT3nA/www.bioverlay.org/post/2019-03-tracking-the-popularity-and-outcomes-of-all-biorxiv-preprints/

We agree that the impact of publication on preprint downloads is an interesting line of inquiry—we have added a citation to Dr Kramer's analysis (https://tinyurl.com/rxivist-further-analysis), which is referenced at the end of the Results section that deals with Figure 5. The paragraph now reads (changes in italics):

“*It is important to note that w*e did not evaluate when these downloads occurred, relative to a preprint's publication: While it looks like accruing more downloads makes it more likely that a preprint will appear in a higher impact journal, it is also possible that appearance in particular journals drives bioRxiv downloads after publication. *The Rxivist dataset has already been used to begin evaluating questions like this (Kramer 2019), and further study may be able to unravel the links, if any, between downloads and journals.*”

10) An automatically generated visualization of the database schema in the methods would be helpful to reusers.

An illustration of the schema (and its relevant inter-table references) has been added to Supplementary File 1, which already includes detailed descriptions of the fields in each table. We also added a legend for the figure, for those unfamiliar with the visualization.

Other feedback from the biOverlay.org reviews:

Reviewer 1 (Dr Daniel Himmelstein):

Another interesting finding reported in the Methods — which also points to the satisfactory quality & rigor of existing preprints — was that: "A search for 1,345 journals based on the list compiled by Stop Predatory Journals showed that bioRxiv lists zero papers appearing in those publications." I think one explanation is that early preprint adopters were generally forward-thinking, competent, perceptive, and attuned researchers. Such researchers are unlikely to publish in predatory journals. However, another possibility is that bioRxiv is not detecting publications in predatory journals, as described in my following comment.

Response: We agree that there is more than one possible explanation for the absence of noted predatory journals in the bioRxiv data linking preprints to the publications in which they appeared. A sentence has been added to the Methods section (page 40) to clarify our intentions in performing this check—we did not mean to imply that preprints have never appeared in a predatory journal, only that those journals were not factored into our analysis of publication rates (new text in italics):

A search for 1,345 journals based on the list compiled by Stop Predatory Journals (https://predatoryjournals.com) showed that bioRxiv publication data did not include any instances of papers appearing in those *journals* ("List of Predatory Journals" 2018). *It is important to note that the absence of this information does not necessarily indicate that preprints have not appeared in these journals—we performed this search to ensure our analysis of publication rates was not inflated with numbers from illegitimate publications.*

The authors might also consider using “Impact Factor” rather than “impact score”, since the JIF is just one of many scores that can be used to assess a journal (and is known to be a methodologically deficient measure of mean citations).

Response: To clarify the specific journal impact metric we used in our analysis, references to "impact score" have been changed throughout the manuscript to refer instead to "Journal Impact Factor" or "JIF."

I could not find the manuscript figures (or text) anywhere besides the PDF. Therefore, I had to take screenshots to extract the figures for my journal club presentation. It would be useful to add the figures to one of the existing repositories or data depositions.

Response: PDF files for each figure have been added to the data deposition associated with the manuscript, available at https://doi.org/10.5281/zenodo.2603083

Reviewer 2 (Dr Devang Mehta):

… I do however think a live tally of the more global data that the authors present (number of downloads by field, submissions etc.) updated daily on the site would’ve been a useful feature to have on the website.

Response: A feature to display summary statistics (overall downloads, submissions, etc.) has been added to the website, and is now available at https://www.rxivist.org/stats.